# Far-Ultraviolet Light at 222 nm Affects Membrane Integrity in Monolayered DLD1 Colon Cancer Cells

**DOI:** 10.3390/ijms25137051

**Published:** 2024-06-27

**Authors:** Jun Nishikawa, Yuta Tamura, Tomohiro Fujii, Soichiro Fukuda, Shoma Yoneda, Nanami Yamaura, Shinichi Takahashi, Takeshi Yamamoto, Junzo Nojima, Yutaka Suehiro, Takahiro Yamasaki, Taro Takami

**Affiliations:** 1Faculty of Laboratory Science, Yamaguchi University Graduate School of Medicine, Ube 755-8505, Japan; c007upw@yamaguchi-u.ac.jp (Y.T.); tomo329@yamaguchi-u.ac.jp (T.F.); fukuda@yamaguchi-u.ac.jp (S.F.); c012upw@yamaguchi-u.ac.jp (S.Y.); d013upu@yamaguchi-u.ac.jp (N.Y.); d009upu@yamaguchi-u.ac.jp (S.T.); kenyama@yamaguchi-u.ac.jp (T.Y.); nojima-j@yamaguchi-u.ac.jp (J.N.); 2Department of Oncology and Laboratory Medicine, Yamaguchi University Graduate School of Medicine, Ube 755-8505, Japan; ysuehiro@yamaguchi-u.ac.jp (Y.S.); t.yama@yamaguchi-u.ac.jp (T.Y.); 3Department of Gastroenterology and Hepatology, Yamaguchi University Graduate School of Medicine, Ube 755-8505, Japan; t-takami@yamaguchi-u.ac.jp

**Keywords:** far-ultraviolet, 222 nm, membrane damage, monolayer cells, cell death

## Abstract

222 nm far-ultraviolet (F-UV) light has a bactericidal effect similar to deep-ultraviolet (D-UV) light of about a 260 nm wavelength. The cytotoxic effect of 222 nm F-UV has not been fully investigated. DLD-1 cells were cultured in a monolayer and irradiated with 222 nm F-UV or 254 nm D-UV. The cytotoxicity of the two different wavelengths of UV light was compared. Changes in cell morphology after F-UV irradiation were observed by time-lapse imaging. Differences in the staining images of DNA-binding agents Syto9 and propidium iodide (PI) and the amount of cyclobutane pyrimidine dimer (CPD) were examined after UV irradiation. F-UV was cytotoxic to the monolayer culture of DLD-1 cells in a radiant energy-dependent manner. When radiant energy was set to 30 mJ/cm^2^, F-UV and D-UV showed comparable cytotoxicity. DLD-1 cells began to expand immediately after 222 nm F-UV light irradiation, and many cells incorporated PI; in contrast, PI uptake was at a low level after D-UV irradiation. The amount of CPD, an indicator of DNA damage, was higher in cells irradiated with D-UV than in cells irradiated with F-UV. This study proved that D-UV induced apoptosis from DNA damage, whereas F-UV affected membrane integrity in monolayer cells.

## 1. Introduction

A light source device that can selectively irradiate with far-ultraviolet (F-UV) light at a wavelength of 222 nm has been developed by combining an excimer lamp and a filter [1]. This F-UV light of 222 nm wavelength has a bactericidal effect similar to that of deep-ultraviolet (D-UV) light of about a 260 nm wavelength [2,3]. It can sterilize drug-resistant bacteria such as methicillin-resistant *Staphylococcus aureus* and extended-spectrum β-lactamase-producing bacteria [4,5] and inactivate several viruses, including SARS-CoV-2 [6].

Because the wavelength of F-UV is very short, the light does not penetrate deep into the skin or the eyes. Kaidzu et al. reported that the dose that injures the rat cornea is between 3500 and 5000 mJ/cm^2^ [7]. In a study of 222 nm F-UV irradiation of 50–500 mJ/cm^2^ delivered to the backs of 20 healthy volunteers, no skin erythema was observed [8]. Thus, F-UV irradiation at 222 nm could be safe for the skin and cornea, in contrast to D-UV irradiation at 260 nm [9]. Threshold limit values for 222 nm F-UV recommended by the American Conference of Governmental Industrial Hygienists were recently revised to 160 mJ/cm^2^ for the eye and to 479 mJ/cm^2^ for skin [10]. The F-UV irradiator has begun to be used in various environments, including a shared bathroom [11].

The cytotoxicity of D-UV is due to DNA damage because 260 nm is the absorption maximum of DNA [12,13]. F-UV radiation is thought to be less cytotoxic and less harmful to the human body, but the extent of this effect has not been fully investigated. In this study, we irradiated monolayer cells with F-UV light to investigate the effects of F-UV light on cells.

## 2. Results

### 2.1. Cell Viability and Morphologic Changes

When monolayer-cultured DLD-1 cells were irradiated with UV light, F-UV and D-UV showed cytotoxicity in a radiant energy-dependent manner (Figure 1a). The viable cell rate decreased to 24.9 ± 1.54% at 24 h and 10.7 ± 1.48% at 48 h after 30 mJ/cm^2^ of F-UV irradiation. Similarly, irradiation of DLD-1 cells with 30 mJ/cm^2^ of D-UV light reduced the viability rate to 34.8 ± 5.83% after 24 h and to 17.5 ± 3.06% after 48 h (Figure 1b). Irradiation energy of 30 mJ/cm^2^ was used in the following experiments because F-UV and D-UV showed comparable cytotoxicity.

As shown by phase contrast microscopy, the DLD-1 cells were cultured in a monolayer before UV irradiation. After 30 mJ/cm^2^ F-UV irradiation, the DLD-1 cells gradually began to expand, and many cells had expanded after 4 h of irradiation (Appendix A, Figure 2a). In contrast, the D-UV-irradiated cells did not expand with 4 h of irradiation (Figure 2a). DAPI staining showed little nuclear change in the F-UV-irradiated cells, whereas D-UV irradiation resulted in nuclear aggregation and fragmentation and cell shrinkage (Figure 2a).

### 2.2. Cyclobutane Pyrimidine Dimer Production

DNA was extracted from cells irradiated with UV light, and the amount of CPD was measured using ELISA. The amount of CPD was 3.37 ± 0.02 ng/µg DNA for F-UV irradiation and 13.1 ± 1.62 ng/µg DNA for D-UV irradiation, indicating severe DNA damage with D-UV irradiation (Figure 2b). No statistically significant difference was found between F-UV and D-UV irradiation.

### 2.3. Live/Death Assay

DLD-1 cells immediately after F-UV irradiation were stained only with Syto9, and the cells showed marked PI uptake at 4, 24 and 48 h later (Figure 3). Contrastingly, after D-UV irradiation, cells were stained only with syto9, and PI was not incorporated into the cells for 48 h (Figure 3). PI uptake reflects plasma membrane damage because PI is a DNA-binding fluorescent dye that is membrane-impermeable. As PI binds to DNA more strongly than syto9, cells in which PI is incorporated are known to have attenuated syto9 signaling.

## 3. Discussion

Because of its short wavelength, F-UV radiation generally does not penetrate deeply into tissues. Hanamura et al. evaluated the cell viability of layered cell sheets by irradiation with 222 nm F-UV light and showed that F-UV was transmitted less to lower layers and affected the viability of the cells less than irradiation at 254 nm D-UV [14]. When we cultured DLD-1 cells in a monolayer and irradiated them with 222 nm F-UV at 30 mJ/cm^2^, cell death was induced, and we investigated the mechanism of cell death by F-UV in this study.

DLD-1 cells began to expand immediately after 222 nm F-UV light irradiation, and many cells incorporated PI. In contrast, cells irradiated with D-UV light at 254 nm shrank slightly after irradiation, and the nuclei aggregated, indicating apoptosis. In addition, the amount of CPD, an indicator of DNA damage, was higher in cells irradiated with D-UV than in cells irradiated with F-UV. This study proved that D-UV induced apoptosis from DNA damage. D-UV irradiation causes the accumulation of DNA damage over time; DNA damage can be repaired, but incomplete repair or accumulation of damage leads to cell death. Therefore, we believe that cell viability decreases with time. F-UV, on the other hand, affected the cell membrane of monolayer cells and induced cell death (Figure 4). Kang et al. examined membrane integrity using a PI uptake assay following F-UV irradiation of some bacteria and showed that the degree of PI uptake by 222 nm F-UV-treated bacteria was much higher than that by 254 nm D-UV-irradiated bacteria [2]. They showed that 222 nm F-UV induced the generation of reactive oxygen species (ROS), which resulted in lipid peroxidation in the bacterial membrane [2]. In our experiments, dichlorofluorescein, which is oxidized and fluoresces in the presence of ROS, was examined, but no fluorescence was observed in the DLD-1 cells after D-UV or F-UV irradiation. The mechanism of cell membrane damage by 222 nm F-UV is a future issue to be investigated.

A spectrophotometric analysis showed that 222 nm F-UV light is absorbed more by proteins than D-UV light [15], and 260 nm is the absorption maximum of DNA. Wavelength-dependent photobiochemical mechanisms were investigated with 222 and 254 nm irradiation by Naito et al. The ability of plasmid DNA to transform Escherichia coli was greatly reduced in the D-UV-irradiated plasmids compared with the F-UV-irradiated plasmids [16], whereas protease activity was significantly reduced by 222 nm F-UV irradiation compared with 254 nm irradiation [16]. Therefore, cytoplasmic proteins, such as protease, and not nuclear DNA, are one of the candidates targeted by F-UV.

F-UV is a light source known for its effects on SARS-CoV-2. When a solution of SARS-CoV-2 was irradiated with 222 nm F-UV, the viruses lost their infectivity, but no difference was observed in the content of the nucleic acids detected by quantitative PCR [6], suggesting that the envelope protein of the virus may be destroyed. Oh et al. reported that 222 nm F-UV efficiently damaged the adenovirus capsid protein [17]. These results may also indicate that F-UV light targets membranous proteins rather than nucleic acids.

Near-infrared light has shown promise in cancer therapy due to its deeper tissue penetration and minimal damage to healthy tissues [18]. In addition, near-infrared photoimmunotherapy has emerged as a targeted cancer treatment that induces immunogenic cell death [19]. F-UV is reportedly less harmful to the skin and eyes than D-UV because its short wavelength does not penetrate deep into the human tissue. Because F-UV was as cytotoxic as D-UV to monolayer cultured cells in the present study, repeated and selective F-UV irradiation may induce necrosis in early stage tumor cells without damaging the normal epithelium. Currently, we consider that F-UV is far from being applicable to the treatment of colorectal cancer, but we would like to report in this study that the mechanism-induced cell death by F-UV is different from those of D-UV radiation (Figure 4).

Our primary interest is to investigate the mechanisms by which F-UV radiation affects cell membranes and induces necrosis through membrane damage. Specifically, we intend to examine changes in the localization of cell adhesion factors on the plasma membrane, changes in paracellular and transcellular permeability, and changes in plasma membrane morphology as observed by electron microscopy. In addition, the investigation of target proteins in the cell membrane or cytoplasm of F-UV-induced cell death will be a concern for future studies.

## 4. Materials and Methods

### 4.1. Cell Culture and Far-Ultraviolet Irradiation

Colon cancer cell line DLD-1 was cultured in RPMI 1640 supplemented with 10% FBS and 1% penicillin–streptomycin. Then, 1.0 × 10^4^ DLD-1 cells were seeded in 96-well plates and incubated at 37 °C for 48 h. After removing the medium, the cells were washed twice with 100 µL of PBS. After removal of PBS, the cells were irradiated with 222 nm of F-UV or 254 nm of D-UV. Then, 100 µL of culture medium was added, and the culture was continued at 37 °C in 5% CO_2_ until assays were performed.

A filtered Kr-Cl F-UV source (Care222™, Ushio Inc., Tokyo, Japan) with peak wavelength of 222 nm was used. The radiant energy at 100 mm from the F-UV source to the cell surface was 0.974 (≈1) mW/cm^2^. We performed time series tests with irradiation times of 0, 10, 30, 60, 180, and 300 s. Thus, the F-UV radiant energy at each irradiation time was 0, 10, 30, 60, 180, and 300 mJ/cm^2^, respectively. The device used to emit D-UV at a wavelength of 254 nm was an SLUV-4 lamp (Azwan Corporation, Osaka, Japan). The radiant energy at 50 mm from the D-UV source was 0.614 mW/cm^2^, according to the instruction manual. Therefore, the D-UV radiant energy for each irradiation time was 0, 6.1, 18.4, 36.8, 110.5, and 184.2 mJ/cm^2^, respectively. The viability of DLD-1 cells at 48 h after UV irradiation was measured.

According to the results of the time series test, the F-UV and D-UV radiant energies were then unified at 30 mJ/cm^2^, and the percentage of viable cells was measured after 4, 24, and 48 h of UV irradiation.

### 4.2. Cell Viability Assay

Cell viability was assessed by 3-(4, 5-dimethylthiazol-2-yl)-5-(3-carboxymethoxyphenyl)-2-(4-sulfophenyl)-2H tetrazolium (MTS) (Promega) assay. The stock concentration for MTS is 1.90 mg/mL, and the final concentration is 317 µg/mL. After irradiation of F-UV or D-UV, 20 µL of MTS reagent was added, and cells were incubated at 37 °C for 1 h. From this culture, 100 µL of solution was transferred to each 96-well plate for measurement, and the absorbance at 490 nm was measured using a microplate photometer (MultiSkan FC, Thermo Fisher Scientific Corporation, Waltham, MA, USA). Control means the sample was not exposed to UV, whereas the sample designated as 0 h was assayed immediately after UV irradiation. Although the time was not measured, the data were obtained within 30 min after irradiation. The absorbance of the cells at the time of irradiation was used as the standard, and the absorbance of the cells after irradiation was expressed as a percentage defined as the percentage of viable cells. The experiment was performed 3 times, and the average percentage of viable DLD1 cells was evaluated.

### 4.3. Time-Lapse Imaging

For time-lapse imaging, 3 × 10^4^ DLD-1 cells were seeded in a 35 mm dish and incubated at 37 °C for 48 h. The supernatant was then removed, and the cells were washed twice with 1 mL of PBS. The PBS was removed, and the cells were irradiated with 30 mJ/cm^2^ of 222 nm of F-UV or 254 nm of D-UV. After irradiation, 2 mL of culture medium was added. The cells were time-lapse imaged for 48 h, and the video was reduced to 2 min. A C-mount CMOS camera with an IMX322 (SONY) sensor and original small CO^2^ incubation chamber were installed on the Nikon Diaphot inverted microscope. The original magnification was 10×. An image of the cells was captured every 15 min by the original capture software equipped with stepper motor focus control.

### 4.4. Cyclobutane Pyrimidine Dimer (CPD) Production

DNA was extracted from cells after 48 h of exposure to 30 mJ/cm^2^ of 222 nm F-UV or 254 nm D-UV using an All Prep DNA/RNA Mini Kit (Qiagen). ELISA was performed using an OxiSelect™ UV-Induced DNA Damage ELISA Kit (CELL BIOLABS Inc., San Diego, CA, USA). Absorbance at 450 nm was measured by a microplate photometer (MultiSkan FC, ThermoFisher Scientific Corporation, Waltham, MA, USA) to determine the amount of CPD produced. The experiment was performed 3 times, and the average amount of CPD was evaluated.

### 4.5. Live/Death Assay

A staining solution was prepared by adding 1 μL of syto9 and 2 μL of propidium iodide (PI) to 1 mL of RPMI 1640 medium, and 100 µL of this solution was added to the cells after 30 mJ/cm_2_ of 222 nm of F-UV or 254 nm of D-UV irradiation. The stock concentration for Syto9 is 5 mM, and the final concentration is 5 µM. The stock concentration for PI is 1.5 mM, and the final concentration is 3 µM. The cells were shielded from light and incubated at 37 °C for 30 min. The staining solution was then removed by washing with 100 µL of PBS, and 100 µL of culture medium was added for observation with a BZ-X700 all-in-one fluorescence microscope (Keyence, Osaka, Japan). The cells were examined immediately after UV irradiation and at 4, 24, and 48 h after irradiation.

### 4.6. Statistical Analysis

Data are presented as the mean ± standard error. The results of F-UV and D-UV irradiation were statistically compared using the Student *t*-test. A value of *p* < 0.05 was considered to indicate a statistically significant difference.

## 5. Conclusions

We showed that F-UV light induces cell death in monolayered cells and that membrane damage is greater than DNA damage as a target of F-UV light action.

## Figures and Tables

**Figure 1 ijms-25-07051-f001:**
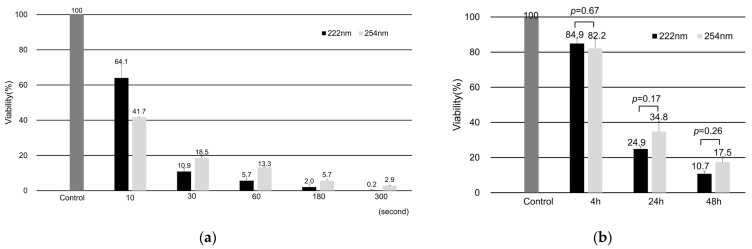
Cell viability of UV-irradiated DLD-1 cells. (**a**) Time series tests with UV irradiation times of 0, 10, 30, 60, 180, and 300 s. Both far-ultraviolet (F-UV) and deep-ultraviolet (D-UV) irradiation showed cytotoxicity as irradiation time increased at 48 h after UV irradiation. (**b**) The viability of DLD-1 cells at 4, 24, and 48 h after 30 mJ/cm^2^ UV irradiation. The absorbance of MTS is divided by the absorbance before irradiation (control), expressed as a percentage. Mean values with standard errors are shown in the graphs.

**Figure 2 ijms-25-07051-f002:**
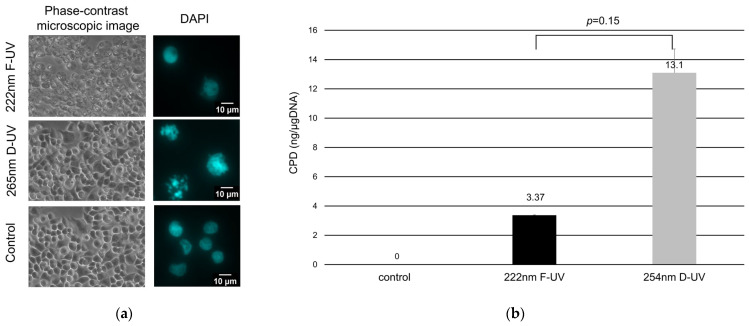
Morphologic changes and cyclobutane pyrimidine dimer (CPD) production of UV-irradiated DLD-1 cells. (**a**) The images 4 h after UV irradiation and the control via phase contrast microscopy and the nucleus of the DLD-1 cells irradiated with UV are shown using DAPI staining. (**b**) DNA samples extracted after F-UV irradiation (black bars) or D-UV (grey bars) were measured for amounts of CPD per µg of DNA. ELISA for CPD was the result of three wells derived from the same samples. Mean values with standard errors are shown in the graphs.

**Figure 3 ijms-25-07051-f003:**
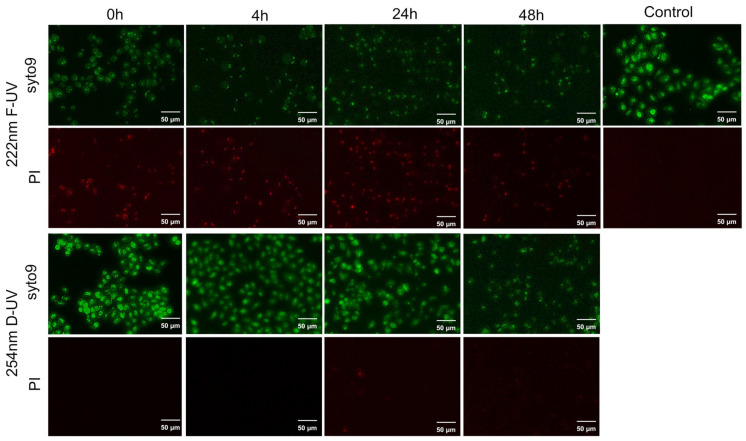
Live/death assay of UV-irradiated DLD-1 cells. DLD-1 cells after far-ultraviolet (F-UV) or deep-ultraviolet (D-UV) irradiation.

**Figure 4 ijms-25-07051-f004:**
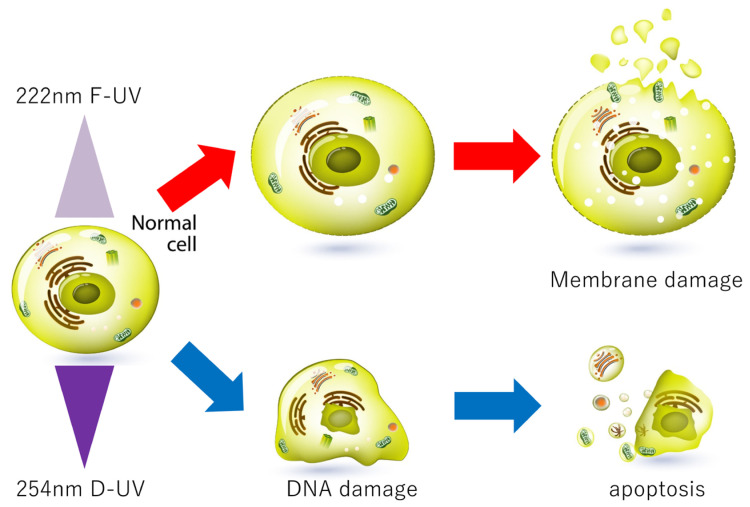
A Graphic summary of the study.

## Data Availability

The data presented in this study are available upon request from the corresponding author.

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
