# Peer review of "Far-Ultraviolet Light at 222 nm Affects Membrane Integrity in Monolayered DLD1 Colon Cancer Cells"

_ijms, 2024, doi:10.3390/ijms25137051_

Round 1
Reviewer 1 Report
Comments and Suggestions for Authors
Good.
Author Response
Thank you for reviewing our manuscript. We have carefully reviewed the comments from you and have revised the manuscript accordingly.
Comments
- Figure 1b shows the cell viability decrease with time, explain mechanism.
(Answer)
UV irradiation causes accumulation of DNA damage over time; DNA damage can be repaired, but incomplete repair or accumulation of damage leads to cell death. Therefore, we believe that cell viability decreases with time. We also consider that F-UV radiation induces cell death by mechanisms other than DNA damage. We have included this in the Discussion section.
- Fig 3 Live/dead assay 254nm D-UV control data is missing. What is the difference between 0h and control? Please clear the confusion.
(Answer)
We apologize for the confusing wording. Control means the sample was not exposed to UV, thus only one image is shown, whereas the sample designated as 0 h was assayed immediately after UV irradiation. Although the time was not measured, the data was obtained within 30 minutes after irradiation. We made the correction in the Material and Methods section.
- There are many evidence that UV radiation have limitation, so researcher is using IR radiation instead UV. Why your F-UV is better?
(Answer)
As you point out, we agree that UV radiation has limitations, and researchers are using IR radiation instead of UV. Several references were cited and added to the Discussion.
F-UV is reportedly less harmful to the skin and eyes than D-UV because its short wavelength does not penetrate deep into the human tissue. This is the good point about F-UV radiation, unlike D-UV radiation. Because F-UV was as cytotoxic as D-UV to monolayer cultured cells in the present study, repeated and selective F-UV irradiation may induce necrosis in early-stage tumor cells without damaging the normal epithelium. Currently, we consider that F-UV is far from being applicable to the treatment of colorectal cancer, but we would like to report in this study that the cytotoxicity of F-UV radiation on cells in a monolayer culture and its mechanism are different from those of D-UV radiation.
- Deep penetration irritation is always creating more damage for tumor. Why the author thinks Far one is better. Is there any radiation data or flow apoptosis data for comparison?
(Answer)
As you point out, we agree that longer wavelength light is more effective because deep effects on tumors are expected. Several references were cited and added to the Discussion. Please refer to our response to your Comment 3 above.
The present study is the first report comparing the effects of F-UV irradiation on cells with those of D-UV irradiation. We found that apoptosis was observed in D-UV irradiated cells, but not in F-UV irradiated cells, based on changes in nuclear morphology after UV irradiation. In addition, CPD, a marker of DNA damage, was also higher in the D-UV irradiated cells, suggesting that D-UV targets nuclear DNA, whereas F-UV causes cells to swell, allowing PI to be taken up. Therefore, the F-UV target was considered to be the cell membrane. A graphic figure has been added to explain this discussion.
- There is talk about ROS, where is the data providing the evidence for it.
(Answer)
Dichlorofluorescein (DCF), which is oxidized and fluoresces in the presence of ROS, was examined, but no fluorescence was observed in the cells after D-UV or F-UV irradiation. The data have not shown as a result.
- There are many data missing, this all result not provide enough evidence for this research.
(Answer)
As you point out, many necessary data for this study are currently missing. Our primary interest lies in exploring the mechanisms by which far ultraviolet (UV) radiation affects cell membranes and induces necrosis through membrane damage. Specifically, we intend to investigate:
- Changes in the localization of cell adhesion factors on the plasma membrane
- Alterations in paracellular and transcellular permeability
- Modifications in plasma membrane morphology observed via electron microscopy
However, we do not yet have data available to present on these aspects, which will be the focus of future research. Consequently, we have included a description of the study's limitations in the discussion section.
Reviewer 2 Report
Comments and Suggestions for Authors
After a careful reading of the manuscript entitled “Far-ultraviolet light at 222 nm affects membrane integrity in monolayered DLD1 colon cancer cells” I have the following comments:
1- Section Results, Cell viability and morphologic changes, the time series presented in Figure 1a (10, 30, 60, 180, 300) is not matched with what is written in Figure 1 legend and Materials and Methods section (10, 20, 30, 60, 180). Please check
2- Figure 1a is missing the statistical analysis and please indicate the time at which you determine the cell viability.
3- Figure 2a, 4 h phase contrast image of F-UV shows that the cells became rounded and started to die as shown in Figure 3 they incorporate more PI. However, the cells of D-UV seem healthier and similar to control (Figure 2a) and they did not incorporate PI (Figure 3). I wonder how different mechanisms of action could have the same impact on cell viability (Figure 1b at 4h). In addition, why authors did not determine the CPD at 4 h?
4- Materials and Methods section, please correct “CO2” “ ≒1”
5- Materials and Methods section, cell viability assay, please indicate the time at which you preformed the MTS assay.
6- Materials and Methods section, what are the stock and final concentrations of MTS, syto9, PI?
7- Authors indicated in Materials and Methods section that they washed the cells with PBS followed by irradiation and then addition of culture medium. Why authors irradiated the cells in the absence of culture medium?
Author Response
For reviewer 1,
Thank you for reviewing our manuscritpt. We have carefully reviewed the comments from you
and have revised the manuscript accordingly.
Comments
After a careful reading of the manuscript entitled “Far-ultraviolet light at 222 nm affects membrane integrity in monolayered DLD1 colon cancer cells” I have the following comments:
1- Section Results, Cell viability and morphologic changes, the time series presented in Figure 1a (10, 30, 60, 180, 300) is not matched with what is written in Figure 1 legend and Materials and Methods section (10, 20, 30, 60, 180). Please check
(Answer)
Thank you for noting this. I have corrected the wording in the legend.
2- Figure 1a is missing the statistical analysis and please indicate the time at which you determine the cell viability.
(Answer)
In the experiment in Figure 1a, the irradiation time of the far UV and deep UV was unified, so the irradiation energies of the two UVs were different. Therefore, we did not perform a statistical analysis between them, however, the results of the statistical analysis were added to Figure 1a. In this study, the percentage of viable cells was measured at 48 hours. This is indicated in the legend.
3- Figure 2a, 4 h phase contrast image of F-UV shows that the cells became rounded and started to die as shown in Figure 3 they incorporate more PI. However, the cells of D-UV seem healthier and similar to control (Figure 2a) and they did not incorporate PI (Figure 3). I wonder how different mechanisms of action could have the same impact on cell viability (Figure 1b at 4h). In addition, why authors did not determine the CPD at 4 h?
(Answer)
We believe that D-UV and F-UV induce cell death by different mechanisms. D-UV induces apoptosis through DNA damage. In contrast, F-UV irradiation induces PI uptake with little CPD and no nuclear fragmentation. Therefore, we suggested that F-UV radiation induces necrosis by damaging cell membranes. A graphic figure has been added to explain this discussion as Figure 4.
As you indicated, CPD should have been measured after 4 hours of UV exposure, but this was not done. Based on the viable cell percentage results, it was decided that the CPD should be measured when sufficient cell death had occurred, which was measured at 48 hours. At 48 hours, the viable cell percentage had decreased to 10.7% for F-UV and 17.5% for D-UV.
4- Materials and Methods section, please correct “CO2” “ ≒1”
(Answer)
Thank you for pointing this out. I have made the correction.
5- Materials and Methods section, cell viability assay, please indicate the time at which you preformed the MTS assay.
(Answer)
In the experiment in Figure 1a, the percentage of viable cells was measured at 48 hours. In the experiment in Figure 1b, the percentage of viable cells was measured at 4 h, 24 h, and 48 h. This is described in the Materials and Methods section.
6- Materials and Methods section, what are the stock and final concentrations of MTS, syto9, PI?
(Answer)
The stock concentration for MTS is 1.90 mg/mL, and the final concentration is 317 µg/mL. The stock concentration for Syto9 is 5 mM, and the final concentration is 5 µM. The stock concentration for PI is 1.5 mM, and the final concentration is 3 µM.
We have added these values to the Materials and Methods section.
7- Authors indicated in Materials and Methods section that they washed the cells with PBS followed by irradiation and then addition of culture medium. Why authors irradiated the cells in the absence of culture medium?
(Answer)
Although F-UV light is not absorbed by water, we were concerned that it would be absorbed by culture medium supplemented with phenol red. Thus, UV light was applied immediately after washing the cells with PBS.
Round 2
Reviewer 1 Report
Comments and Suggestions for Authors
ok. Accept it in its present form.
Comments on the Quality of English LanguageIts seems fine.
Reviewer 2 Report
Comments and Suggestions for Authors
Thanks for your answer.